# Semaphorin Signaling in Cancer-Associated Inflammation

**DOI:** 10.3390/ijms20020377

**Published:** 2019-01-17

**Authors:** Giulia Franzolin, Luca Tamagnone

**Affiliations:** 1Department of Oncology, University of Torino Medical School, 10100 Turin, Italy; giulia.franzolin@ircc.it; 2Cancer Cell Biology Laboratory, Candiolo Cancer Institute-FPO, IRCCS, 10060 Candiolo, Italy; 3Istituto di Istologia ed Embriologia, Università Cattolica del Sacro Cuore, 10168 Rome, Italy; 4Fondazione Policlinico Universitario Agostino Gemelli, 10168 Rome, Italy

**Keywords:** semaphorin, plexin, neuropilin, signaling, cancer, inflammation, immunity, lymphocyte, macrophage

## Abstract

The inflammatory and immune response elicited by the growth of cancer cells is a major element conditioning the tumor microenvironment, impinging on disease progression and patients’ prognosis. Semaphorin receptors are widely expressed in inflammatory cells, and their ligands are provided by tumor cells, featuring an intense signaling cross-talk at local and systemic levels. Moreover, diverse semaphorins control both cells of the innate and the antigen-specific immunity. Notably, semaphorin signals acting as inhibitors of anti-cancer immune response are often dysregulated in human tumors, and may represent potential therapeutic targets. In this mini-review, we provide a survey of the best known semaphorin regulators of inflammatory and immune cells, and discuss their functional impact in the tumor microenvironment.

## 1. Cancer-Associated Immune Response and Semaphorin Signals

It is widely accepted that chronic inflammation predisposes to cancer development. Moreover, there is always a remarkable representation of inflammatory cells in the tumor microenvironment (TME), which is probably elicited by consequences of the aberrant neoplastic growth, such as tissue hypoxia and cell necrosis. This cancer-associated inflammation is characterized by the recruitment of white blood cells and by the activity of inflammatory cytokines and chemokines, as well as by the associated regulation of angiogenesis, fibrosis, and tissue remodeling [1,2]. In fact, the tumor microenvironment may comprise different kinds of inflammatory cells: from those basically implicated in the innate immune response, such as neutrophils and monocytes/macrophages, to cells implicated in mounting the adaptive immunity, such as dendritic cells and lymphocytes. The function of these cells is actually deployed in aberrant manner in the tumor microenvironment, compared to other inflamed tissues. For instance, Natural Killer (NK) cells do not seem to be efficiently mediating cancer cell killing, and the rise of antigen-specific immune response seems to be hindered at various steps [3].

In fact, many inflammatory cells found in the tumor microenvironment seem to act as inhibitors, rather than promoters of the immune response [4]. For example, tumor-associated macrophages (TAMs) and myeloid-derived suppressor cells (MDSCs) recruited from the bone marrow are educated by cancer cells to exert pro-tumorigenic functions, such as sustaining tumor growth, immunosuppression, angiogenesis, and metastatic dissemination [5,6,7]. This alternative phenotype of TAMs is often indicated as M2, by contrast to M1-differentiation state, which is more typical of inflammation elicited by external pathogens. M2-type macrophages are also found in tissues affected by chronic inflammation, and are considered potential drivers of the development of tumor foci in this context [8].

Notably, even if cancer cells often express aberrant proteins (e.g., due to genomic mutations), the antigen-specific adaptive immune response is also hindered in the tumor microenvironment, as soluble cytokines prevent T cell infiltration and cell surface signals are tuned to inhibit lymphocyte-mediated killing activity [4]. In addition, a subset of T regulatory lymphocytes (Treg) can suppress the function of T cytotoxic and T helper cells, the presence of which is instead associated with better prognosis. In fact, therapeutic strategies such as immune checkpoint blockade and immunomodulatory molecules seek to counteract the activity of suppressor cells and promote anti-tumor immune response [9].

Accumulating evidence indicates that diverse members of the Semaphorin family are involved in virtually all phases of physiological and pathological immune responses [10] (Figure 1). Semaphorins actually constitute a large family of conserved proteins, subdivided on the basis of structural and sequence similarity [11,12]. Among mammalian family members, class 3 semaphorins are secreted, whereas classes 4–6 are transmembrane proteins, and semaphorin 7A (Sema7A) is linked to the plasma membrane via a glycophosphatidylinositol (GPI) anchor; moreover, membrane-bound semaphorins can be proteolytically cleaved to generate soluble proteins. Semaphorin signaling is mainly mediated by Plexin and Neuropilin receptors [13]. Plexins comprise nine transmembrane receptors, subdivided into four classes, A–D.

Originally identified as signaling cues for axonal navigation, semaphorins were later found to be involved in the regulation of diverse biological and pathological processes, developmental angiogenesis, bone homeostasis and, indeed, immune response [14]. Some family members can promote the immune response, while others suppress immune cell activation and proliferation. These so-called “immuno”-semaphorins are acquiring increasing importance in the pathogenesis, as well as therapeutic targets, in several immunological diseases such as asthma, psoriasis, and systemic lupus erythematosus [10]. In this context, semaphorin signaling, either provided by cancer cells or by stromal and infiltrating cells in the tumor microenvironment, can skew the inflammatory landscape towards a tumor-promoting or a tumor-suppressive milieu. Moreover, the expression of semaphorins and their receptors is deregulated in tumors, where they can in fact promote or inhibit tumor progression [14,15]. Here, we review the current knowledge on semaphorin signals in cancer-associated inflammation, and discuss the emerging role of semaphorins in tuning the immune response elicited in the tumor microenvironment.

## 2. Semaphorin 4D

Semaphorin 4D (SEMA4D) was originally described in immune cells as CD100 antigen [16] and was the first semaphorin found to have immunoregulatory activity. SEMA4D exists either as a 150-kDa transmembrane full-length isoform, or as a 120-kDa soluble molecule originating from type 1 metalloprotease-mediated proteolytic cleavage of the former. PlexinB1, PlexinB2 and CD72 are the three known receptors for SEMA4D. SEMA4D is broadly expressed in the embryonic nervous tissue and on resting T cells, macrophages, NK cells and neutrophils [17]. The functional receptor for SEMA4D in immune cells is CD72; this is a negative regulator of B cell activation and its signaling cascade is turned off by SEMA4D [18]. Moreover, SEMA4D is required for dendritic cells (DC) activation, in turn regulating T cell functions [19,20].

SEMA4D expression in the tumor microenvironment has been shown to promote tumor progression [21]. Both cancer cells and TAMs have been found to express and release SEMA4D in the tumor microenvironment [21,22]. Notably, TAM-derived SEMA4D is critical for tumor angiogenesis and pericytes recruitment [22]. Moreover, Younis et al. showed that tumor cell-derived SEMA4D could polarize myeloid cells towards MDSC phenotype (CD33^+^,CD11b^+^,HLA-DR^-/low^), with a concomitant reduction in T cell proliferation and IFNγ production [23]. Furthermore, SEMA4D was found to promote the production of immunosuppressive cytokines, whereas its inhibition rescued T cell function, led to increased number of effector T cells and decreased suppressive Tregs. In the same line, Evans et al. demonstrated that SEMA4D expression at the invasive tumor edge creates a barrier to immune infiltration, while antibody-mediated neutralization of SEMA4D could restore the immune response in the TME of murine models of colon cancer and an ERBB2^+^ mammary carcinoma [24]. In particular, SEMA4D-blockade led to a larger number of activated CD86^+^ APCs, CD8^+^ cytotoxic T cells and increased Teffector/Treg ratio, with a concomitant decrease in CD11b^+^ Gr1^+^ MDSCs and in the density of CD206^+^ M2 macrophages. Moreover, the combination with anti-SEMA4D antibodies enhanced the efficacy of immunomodulatory therapies [24].

## 3. Semaphorin 3A

Secreted SEMA3A is probably the best studied semaphorin; its typical receptor complex contains Neuropilin-1 (NRP1) and members of the Plexin-A subfamily. In cancer context, SEMA3A plays pleiotropic activities controlling tumor cells, tumor vessels and infiltrating inflammatory cells. Several data support a vessel-regulating and tumor suppressor function of SEMA3A, but there are discordant findings [25,26]. Indeed, SEMA3A expression was found to decrease in advanced stage tumors, but other studies correlated high tumor levels of SEMA3A with poor outcome [27,28]. In particular, while it has been shown by different groups that SEMA3A can attract NRP1-expressing tumor-infiltrating monocytes, data diverge on the functional role of these cells. For example, upon SEMA3A overexpression in tumor cells, it selectively expanded the population of M1-like pro-inflammatory macrophages derived from NRP1+ monocytes, which further led to the recruitment of NK and CD8+ cytotoxic T cells [29]. However, in murine models deprived of NRP1+ monocytes, tumor growth and metastatic progression were actually inhibited [30]. It was found that SEMA3A can attract tumor-infiltrating macrophages into the hypoxic tumor core, where they are induced to differentiate towards the M2 immune-suppressive and cancer promoting phenotype [30]. These apparent discrepancies probably reflect the multiplicity of SEMA3A-receptor complexes and signaling cascades in different cell types. In particular, the functional role of NRP1 receptor in tumors is very complicated, as it can participate in different signaling complexes [31,32]. Moreover, it was demonstrated that SEMA3A can also have distinctive NRP1-independent activities. For example, SEMA3A acted as inhibitor of the migration of NRP1-deficient myeloid cells, via PlexinA4 [30]. Moreover, recently Gioelli et al. showed that a modified SEMA3A devoid of NRP1-binding can effectively inhibit tumor growth and metastatic dissemination via PlexinA4 [33], expressed in both tumor vessels and immune cells.

In addition to monocytes and TAMs, SEMA3A also regulates T cell function. For instance, SEMA3A signaling has been shown to inhibit T cell activation by eliciting TCR redistribution on the cell surface [34]. Moreover, it was recently shown that SEMA3A can associate with B7-H4, a known inhibitor of TCR-induced proliferation of CD4+ and CD8+ cells; according to this study, SEMA3A mediates B7-H4 immunosuppressive activity by engaging a NRP1/PlexinA4 receptor complex and enhancing the activity of T regulatory cells [35]. However, in other studies, SEMA3A is considered a positive mediator of inflammatory response [36]. Whether SEMA3A can also directly control T cell function in the tumor context awaits clarification; however, as discussed above, its regulatory activity on macrophages and dendritic cells is likely to impact adaptive anti-tumor immune response.

## 4. Semaphorin 4A

SEMA4A is a transmembrane semaphorin, reported to interact with different Plexins, as well as with NRP1, Tim-2, and ILT-4 receptor molecules [37]. In the immune system, SEMA4A is expressed in dendritic cells and B lymphocytes, and it can enhance the activation and differentiation of T cells and the generation of antigen-specific T cells in vivo [38]. In particular, SEMA4A knock-out results in compromised CD8+ T cells activation and differentiation, and impaired in vivo pathogen-specific response [39]. However, while in murine models SEMA4A was shown to induce Th1 immune responses, the human SEMA4A was rather found to elicit robust Th2 responses, with the production of cytokines IL-4, IL-5, and IL-13 [40]. Moreover, another report showed that SEMA4A on the surface of immune cells engaged in trans with NRP1 expressed on Treg lymphocytes, promoting their immunosuppressive activity [41]. SEMA4A also contains an intracellular domain that can mediate reverse signaling cascades into dendritic cells, for instance controlling their migration [42]. The role of SEMA4A in cancer context is still unclear. According to some reports, it could have a promoting effect, possibly acting on inflammatory and immune cells and turning on their activity to foster tumor progression [43].

## 5. Semaphorin 3E

SEMA3E and its receptor PlexinD1 are expressed in both cancer cells and diverse stromal cell populations in the TME. In certain tumors, SEMA3E has been associated with progression due to distinctive signaling cascades promoting cancer cell invasiveness and metastatic spreading [44,45,46]. On the other hand, SEMA3E is also known to restrict vessel development and tumor angiogenesis [47,48]. Even though little is known about the role of SEMA3E in the regulation of tumor-infiltrating immune cells, suggestions for further investigation came from recent reports addressing its role in inflammatory cells [49]. For instance, Wanschel and colleagues showed that SEMA3E mRNA is expressed in macrophages of advanced atherosclerotic plaques in mice and it is highly expressed in inflammatory M1-type, but not anti-inflammatory M2-type macrophages [50]. Moreover, SEMA3E has been found to regulate NK-cell migration, and NK–DC interactions in particular. In fact, mouse NK cells express SEMA3E receptors PlexinD1 and NRP1, and in vitro cell migration assays showed that SEMA3E (released by immature, but not activated DC) inhibits NK cell migration [51]; this could provide a safeguard mechanism preventing NK function in the absence of an active immune response. Moreover, SEMA3E/PlexinD1 signaling was found to negatively regulate the immunological synapse and antigen-specific thymocyte maturation [52,53]. Whether the above mechanisms can also be relevant in response to SEMA3E found in the tumor microenvironment awaits experimental proof.

## 6. Semaphorin 6D

Although SEMA6D expression has been associated with tumor progression and angiogenesis in gastric cancer [54,55], a specific role in the regulation of tumor-associated inflammation has not been investigated. Actually, SEMA6D is a transmembrane semaphorin known to be expressed in lymphocytes, and SEMA6D/PlexinA1 signaling has been implicated in DC-induced T cell function [56], as well as late-stage activation and selection of long-lived lymphocytes [57]. On the other hand, SEMA6D reverse signaling, triggered by PlexinA4 and mediated by Abl kinase associated to its cytoplasmic tail, has been found to reprogram macrophage polarization towards an anti-inflammatory phenotype [58].

## 7. Semaphorin 7A

The GPI-membrane-anchored protein SEMA7A has been found to signal in trans through PlexinC1 and beta-integrins in several systems [59]. Notably, SEMA7A expression is upregulated in breast cancers and is associated with poor overall and distant metastasis-free survival [60], consistent with a pleiotropic tumor-promoting role, potentially including the regulation of inflammatory cells. In the immune system, SEMA7A is expressed on activated T cells and it signals to promote the production of inflammatory cytokines in monocytes and macrophages [61]. Moreover, it has been shown that SEMA7A can prompt TAMs to produce pro-angiogenic molecules such as CXCL2 and MIP-2 [62].

## 8. Overall Conclusion and Future Directions

Cancer-associated immune response is a major player in the tumor microenvironment, controlling cancer cell growth and metastatic dissemination. While many of the implicated mechanisms have not been fully elucidated, the overall picture features the normal function of diverse immune cells being hijacked and distorted by conditioning signals generated by cancer cells. Innate immunity is torn to suppress adaptive response, and instead to promote tumor angiogenesis and cancer progression, by releasing signals typically seen during tissue regeneration. Dendritic cells and lymphocytes are often unable to mount an efficient response against tumor cells, as the latter deploy mechanisms to prevent antigen recognition and suppress cell killing. However, this field is also emerging as a pivotal arena for innovative anti-cancer drugs; for instance, in several cases, immune checkpoint inhibitors succeed in counteracting cancer-induced immune suppressive mechanisms, leading to reinstatement of lymphocyte-mediated tumor cell killing and long-term clearance.

Semaphorins are among major regulatory signals in the tumor microenvironment, and semaphorin receptors are well expressed by all cells involved in the immune response. Indeed, aberrant semaphorin function has been found to sustain autoimmune diseases, and emerging data show the relevance of semaphorin signals in cancer-associated inflammation and immune response. Thus, semaphorins and their receptors are considered novel potential therapeutic targets for the treatment of immune diseases, as well as for re-conditioning the tumor microenvironment in cancer-suppressive, instead of cancer-promoting, mode of action. For instance, the activity of semaphorins controlling M2-type TAMs and Treg lymphocytes could be targeted, such as Sema3A and Sema4D. Future studies are warranted to extensively test this hypothesis in appropriate preclinical models, and eventually transfer putative novel drugs to the clinics.

## Figures and Tables

**Figure 1 ijms-20-00377-f001:**
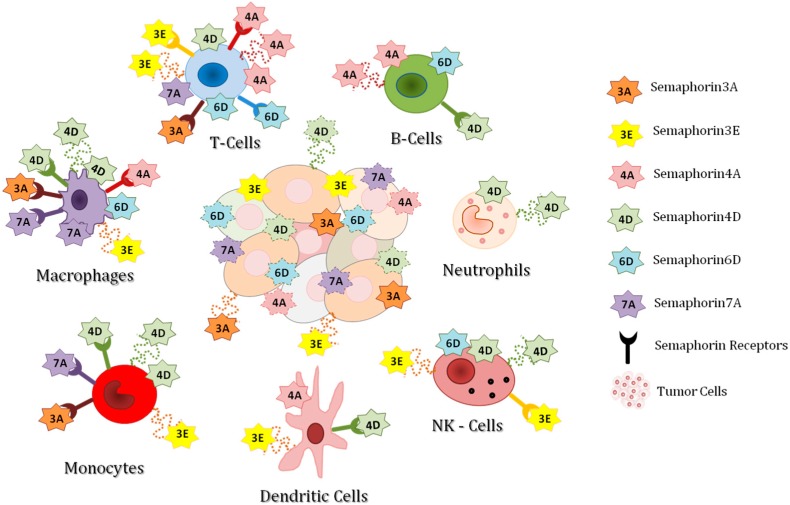
Highlighted in the cartoon are major semaphorin signals potentially relevant for the regulation of diverse inflammatory and immune cells found in the tumor microenvironment. Semaphorins are in fact produced by both cancer cells and cells recruited from the circulation in the inflammatory process. Dotted connecting lines indicate semaphorin molecules secreted or proteolytically shed from the cell surface. Ligand-engaged transmembrane receptors illustrate known autocrine/paracrine signaling mechanisms. Membrane anchored semaphorins might provide juxtacrine proximity signals to neighboring cells (not depicted in the cartoon, for sake of simplicity). A very complex network of signals between different cells in the tumor microenvironment is therefore envisaged. The drawing also summarizes several evidences about semaphorin signaling in normal immune response, the relevance of which in cancer context needs to be assessed. Notably, while for Sema4D there is a consensus on its pro-tumorigenic role, and the multifaceted activity of Sema3A in the tumor microenvironment is under intense investigation, for other immuno-semaphorins, more experimental data are warranted to understand whether they act in cancer-related inflammation as promoters or suppressors.

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
