# Peer review of "Semaphorin Signaling in Cancer-Associated Inflammation"

_ijms, 2019, doi:10.3390/ijms20020377_

Reviewer 1 Report

The authors provide a thorough review of the role of semaphorins in cancer-associated inflammation. Below are comments intended to improve the manuscript: 1) Authors jumped straight into the semaphorins from the introduction. It would be helpful to give a quick primer on cancer-associated immunity prior to discussing the role of each semaphorin. 2) Similarly, after the last semaphorin, there is no clinical application, conclusion, or future directions. Authors should include this type of information at the end. 3) The figure included with the manuscript is not labelled as Figure 1. Nor is it referenced in the text. 4) Figure 1 would be more helpful if the authors could indicate somehow that the semaphorin effects shown are either oncogenic or tumor suppressive.

Author Response

We thank the Reviewer for her/his appreciation of our work and are grateful for the constructive suggestions to improve it. 

Response to specific points below, between lines.

1) Authors jumped straight into the semaphorins from the introduction. It would be helpful to give a quick primer on cancer-associated immunity prior to discussing the role of each semaphorin. 

It was our idea to focus our review on a specific topic and for this reason we had kept to a minimum the Introductory paragraph, both concerning overview of cancer-related inflammation and the semaphorin family, rather providing several references to general Review articles on these topics. In particular we have highlighted the inflammatory/immune cells found to play important roles in the tumor microenvironment; however, we agree with this Reviewer that some additional background information would be useful to the general reader, and for this reason we have added it into the revised version, including one additional reference.

2) Similarly, after the last semaphorin, there is no clinical application, conclusion, or future directions. Authors should include this type of information at the end. 

We fully agree with the Reviewer that a closing section was missing in our original manuscript and thank her/him for this constructive criticism. We have now added a final section containing overall summary and conclusions, and comments on future research directions and prospective clinical relevance.  

3) The figure included with the manuscript is not labelled as Figure 1. Nor is it referenced in the text. 

We apologyze for this mistake, now corrected in the revised manuscript. Thanks for pointing this out.

4) Figure 1 would be more helpful if the authors could indicate somehow that the semaphorin effects shown are either oncogenic or tumor suppressive.

We have tried to modify the figure in order to comply with Reviewer's requests. However, we feel unconfortable with the results, since the field is evolving and for many semaphorins the available data do not allow drawing compelling conclusions about their impact on cancer-related inflammation. Moreover, it should be noted that the cartoon summarizes several evidences about semaphorin signaling also in normal immune response (reported in the manuscript), the relevance of which in cancer context needs to be assessed. As described in the main text, at present, most data about immune-regulation in cancer models concern Sema3A and Sema4D; however, since only for Sema4D a clear consensus exists on its pro-tumorigenic role, we decided to underscore this point in the Figure Legend.

Reviewer 2 Report

The manuscript by Franzolin and Tamagnone on the importance of semaphorin in cancer is well written and describes effects of different known semaphorins in inflammation. The manuscript carefully describes each semaphorin and it's function in inflammation in cancer, however, the manuscript feels inadequate without a proper discussion section summarizing the relevance of the semaphorins and where future research should be directed.

Author Response

We thank the Reviewer for appreciating our work and are grateful for the constructive suggestions to improve it. We fully agree with the Reviewer that a closing section was missing in our original manuscript and thank her/him for this constructive criticism. We have now added a final section containing overall summary, discussion and conclusions, as well as comments on future research directions and prospective clinical relevance.